Accepted at the ICLR 2024 Workshop on AI4Differential Equations In Science

# MultiSTOP: Solving Functional Equations with Reinforcement Learning

**Alessandro Trenta**
University of Pisa
alessandro.trenta@phd.unipi.it

**Davide Bacciu**
University of Pisa
davide.bacciu@unipi.it

**Andrea Cossu**
University of Pisa
andrea.cossu@di.unipi.it

**Pietro Ferrero**
Simon's Center for Geometry and Physics, Stony Brook
pferrero@scgp.stonybrook.edu

## Abstract

We develop *MultiSTOP*, a Reinforcement Learning framework for solving functional equations in physics. This new methodology produces actual numerical solutions instead of bounds on them. We extend the original *BootSTOP* algorithm by adding multiple constraints derived from domain-specific knowledge, even in integral form, to improve the accuracy of the solution. We investigate a particular equation in a one-dimensional Conformal Field Theory.

## 1 Introduction

Functional equations appear in many scientific areas of study due to their ability to efficiently model complex phenomena. Unfortunately, when analytical solutions are not available, finding approximate solutions using numerical methods such as semidefinite programming (Simmons-Duffin, 2015) can often be computationally expensive. Although numerical methods can be quite effective, often they only provide bounds on the parameters of the equation. Differential Equations (DEs) and Partial Differential Equations (PDEs) can also be interpreted as functional equations and have been widely studied with both numerical (Butcher, 2000) and Machine Learning methods (Raissi et al., 2017a;b). We focus on a particular class of equations in the Conformal Field Theories (CFTs) of physics which take the form $h(z, \bar{z}) + \sum_n C_n^2 F_{\Delta_n}(z, \bar{z}) = 0$, with unknowns being the coefficients $C_n^2$ and the functions' parameters $\Delta_n$. Historical approaches (Simmons-Duffin, 2016; Poland et al., 2019) for conformal bootstrap equations involve truncating the equation to all terms with $\Delta_i \leq \Delta_{\max}$ and applying the aforementioned semidefinite programming. With this methodology, only bounds on the unknowns are found (Kos et al., 2014).

Recently, Reinforcement Learning is emerging as a new paradigm to solve functional equations in physics. A recent approach, called Bootstrap STochastic OPtimization, briefly *BootSTOP* (Kàntor et al., 2022; 2023), has been used to find actual numerical solutions for the above equation.

Our main contributions are: (a) MultiSTOP, an extension of BootSTOP which enables the enforcement of additional constraints derived from the physics of the model into the algorithm, in order to increase the accuracy of the solutions. (b) A detailed empirical analysis on the effectiveness of the proposed approach on a $1D$ defect CFT. (c) An analysis of the degeneracy problem that occurs when two or more functions have similar parameters $\Delta_n$ in two limiting cases for the model.

## 2 MultiSTOP

We start with the description of the functional equation we are interested in. We then introduce our MultiSTOP (Multiple STochastic OPtimization) approach solving the conformal bootstrap equation by introducing additional constraints on the unknowns derived from the physical domain.

## 2.1 THE CONFORMAL BOOTSTRAP EQUATION

We focus on the *conformal bootstrap equation* of a Conformal Field Theory (CFT), the $1D$ defect CFT defined by a straight $\frac{1}{2}$-BPS Wilson line in $4D$ $\mathcal{N} = 4$ super Yang-Mills theory (Maldacena, 1998; Drukker & Kawamoto, 2006; Giombi et al., 2017). In this $1D$ setting, the equation reads:

$$h(x) + \sum_n C_n^2 F_{\Delta_n}(x) = 0, \tag{1}$$

and must hold for almost all the points $x \in \mathbb{C}$. Function $h(x)$ is known, while the $F_{\Delta_n}(x)$ are analytical with respect to $\Delta_n$ and are defined in terms of the so called *conformal blocks* $f_{\Delta_n}(x)$ through the formula $F_{\Delta_n}(x) = x^2 f_{\Delta_n}(1 - x) + (1 - x)^2 f_{\Delta_n}(x)$ (see appendix A.1 for a review). Our objective is to find solutions of equation 1 with unknowns given by the squared *OPE coefficients* $C_n^2 \geq 0$ and the *scaling dimensions* $\Delta_n \geq 0$, which we indicate together as *CFT data*. In this model of interest the CFT data depends on a parameter $g$ called *coupling constant*. Previous studies were able to identify the values of the scaling dimensions $\Delta_n$ for the first 10 terms of equation 1 as a function of the coupling constant with high precision for many values of $g$ (see Grabner et al. (2020) and references therein). For our purposes, we give these values as input to the MultiSTOP algorithm, halving the number of unknowns we have to find and significantly reducing the search space.

In the specific model under investigation, the CFT data are further subject to the integral constraints (equation 2), where $\text{RHS}_i$ are functions of $g$ alone, while $\text{Int}_i$ are integrals of the conformal blocks $f_\Delta$ evaluated with respect to a certain measure[1]:

$$\text{Constraint i: } \mathbb{I}_i = \sum_n C_n^2 \text{Int}_i[f_{\Delta_n}] + \text{RHS}_i = 0, \qquad i = 1, 2. \tag{2}$$

These constraints were derived in Cavaglià et al. (2022); Drukker et al. (2022) from the requirement that deformations of the straight line parametrized by the CFT data appearing in equation 1 are compatible with the constraints of supersymmetry, and have been shown to drastically improve the bounds on the numerical determination of the CFT data.

## 2.2 BOOTSTOP

See section A.2 for a brief introduction to the main concepts of Reinforcement Learning (RL).
We start by making two approximations. (a) Since equation 1 consists of an infinite number of terms, we truncate it up to a certain number of terms or to all functions with $\Delta_n \leq \Delta_{\max}$. In this case, we consider the first 10 elements of the sum as their scaling dimensions $\Delta_n$ are known with very high precision (Grabner et al., 2020). For a discussion on how this truncation may affect the results or how to mitigate it through the analysis of the tails see Niarchos et al. (2023). (b) Being equation 1 a functional equation, it has to be verified in a continuous space. In order to make it tractable, we choose a set of 180 testing points in the complex plane where the equation is evaluated. While this does not guarantee the CFT data found to hold for every $x \in \mathbb{C}$, it has proven to be effective with the specific points taken from Kàntor et al. (2023).

We now describe the BootSTOP algorithm, first introduced in Kàntor et al. (2022; 2023), which constitutes the basis for our MultiSTOP approach. In order to apply RL to our framework we need to define a suitable environment with states, actions and, most importantly, a reward signal that guides the agent to find a solution to the equation.
**States**: a state is the current agent's guess of the solution to the truncated version of equation 1, that is the vector of CFT data $S_t = \left(\Delta_1, \ldots, \Delta_{10}, C_1^2, \ldots, C_{10}^2\right) \in [0, \Delta_{\max}]^{10} \times [0, C_{\max}^2]^{10}$. For the $1D$ CFT of interest, it is safe to assume that $\Delta_{\max} = 10$ and $C_{\max}^2 = 1$. We indicate states as $(\boldsymbol{\Delta}, \boldsymbol{C^2})$.
**Actions**: an action by the agent is the change of one couple of values in the current guess $S_t$. In particular, the agent cycles from $n = 1$ to 10 and, at each step, selects the new values for $(\Delta_n, C_n^2)$ obtaining $S_{t+1}$. Note that for this environment the state transitions are deterministic.
**Rewards**: the reward has to guide the agent towards a solution of the conformal bootstrap equation. Evaluating the truncated equation on the aforementioned 180 complex points and with our current guess of the CFT data, we obtain a vector of evaluations $\boldsymbol{E}(\boldsymbol{\Delta}, \boldsymbol{C^2})$. We want this to be as close as possible to the null vector, in order for equation 1 to be satisfied. The closer the norm $\|\boldsymbol{E}(\boldsymbol{\Delta}, \boldsymbol{C^2})\|_2$

---

[1]See appendix A.1 for a more detailed description.

is to 0, the closer the state is to the actual solution and the higher has to be the reward for learning. Possible choices are $R_1 = -\|\boldsymbol{E}(\boldsymbol{\Delta}, \boldsymbol{C^2})\|_2$ as in Kàntor et al. (2022) and $R_2 = \frac{1}{\|\boldsymbol{E}(\boldsymbol{\Delta}, \boldsymbol{C^2})\|_2}$ as in Kàntor et al. (2023). In this work, we consider the second formulation. The RL algorithm used to optimize the agent's policy is Soft Actor-Critic (SAC) (Haarnoja et al., 2018). For the complete algorithm, implementation details and speed-up techniques applied, see appendix A.3.

### 2.3 MULTISTOP: ENFORCING ADDITIONAL PHYSICAL CONSTRAINTS

We now introduce our MultiSTOP, which extends the BootSTOP algorithm by including additional physical constraints. This is an important feature as functional equations or, more generally, DEs or PDEs can feature multiple constraints that must be satisfied, such as initial conditions. Since the physical constraints in equation 2 have proven to be very effective in improving the accuracy of the bounds on the first three values of $C_n^2$ of our model as in Cavaglià et al. (2022), we want to impose these constraints into our RL framework. To do so, we first notice that equation 2 has the same general form and the same unknowns as our objective 1. Hence, we can apply the same reasoning from equation 1: we evaluate the physical constraints on our current guess $(\boldsymbol{\Delta}, \boldsymbol{C^2})$, with a reward that should be higher as $\mathbb{I}_1$ and $\mathbb{I}_2$ in equation 2 are closer to zero. With this in mind, we want to include two terms similar to $\frac{1}{|\mathbb{I}_i|}, i = 1, 2$ into the reward formulation. We have two possibilities:

$$R_1 = \frac{1}{\|\vec{\boldsymbol{E}}_t(\vec{\Delta}, \vec{C^2})\|} + w_1 \frac{1}{|\mathbb{I}_1|} + w_2 \frac{1}{|\mathbb{I}_2|}, \qquad R_2 = \frac{1}{\|\vec{\boldsymbol{E}}_t(\vec{\Delta}, \vec{C^2})\| + w_1|\mathbb{I}_1| + w_2|\mathbb{I}_2|} \qquad (3)$$

where $w_1$ and $w_2$ are weights. After the initial experiments with both forms of reward, with the objective of matching the known bounds for $C_1^2, C_2^2, C_3^2$ with $g = 1$, we found that $R_2$ produces better and more stable results, with weights $w_1 = 10^4$ and $w_2 = 10^5$. While $R_2$ forces all three terms of the denominator to be small to improve the overall reward, when using $R_1$ the agent was sometimes able to optimize one term independently of the others, leading to poor optimization of the total reward.

## 3 RESULTS

We assess the effectiveness of MultiSTOP by studying the $1D$ CFT described in section 2 in two regimes: (a) **weak coupling** ($g \leq 1$), where we investigate the behavior of $C_2^2, C_3^2$ as $g \to 0$ and (b) **Strong coupling** ($g \geq 1$), where we focus on the behavior of $C_n^2, n \geq 4$ as $g \to 4$. In each case, the values for $\Delta_n$ are given as input as they are known with high precision from Grabner et al. (2020). We remark the effectiveness of MultiSTOP with respect to the baseline BootSTOP algorithm: when trying to match the known values for $C_1^2, C_2^3, C_3^2$, integrating the constraints in equation 2 leads to a reduction in relative error from 2x to 10x. This effect has already been observed in similar contexts (Cavaglià et al., 2022; Chester et al., 2022; Chester, 2023). See appendix B.1 and C for further details from a theoretical and experimental point of view.

### 3.1 WEAK COUPLING AND THE PROBLEM OF DEGENERACY

Since the numerical bounds on $C_1^2$ from Cavaglià et al. (2022) are tight, we provided their middle value as input to the algorithm to reduce the search space and help the agent. In figure 1 we plot the results of the best 25 runs based on reward for the tested values of $g$ and compare them with the available bounds in literature. We notice that the results are less coherent when $g \to 0$, but the values for the coefficients are outside of the bounds in a symmetric way. Each value for $C_2^2$ is below the expected ones and the opposite happens for $C_3^2$. If we consider the sum $C_2^2 + C_3^2$, most of the values are in the green regions and therefore acceptable. We found that this is strictly related to the fact that as $g \to 0$, the scaling dimensions $\Delta_2$ and $\Delta_3$ converge to the same value of $\Delta = 2$ (Cavaglià et al., 2022; Grabner et al., 2020). This is known as *degeneracy* and affects the ability of the agent to find accurate values of both the coefficients at the same time (appendix B.2). Since the functions $F_{\Delta_n}(x)$ are analytical with respect to $\Delta_n$, up to terms of $O(\Delta_2 - \Delta_3)$ we can write equation 4, showing that only $C_2^2 + C_3^2$ can be determined with good precision in the limit $\Delta_2 - \Delta_3 \to 0$.

$$C_2^2 F_{\Delta_2} + C_3^2 F_{\Delta_3} \approx (C_2^2 + C_3^2) F_2. \qquad (4)$$

To solve this issue, more conformal bootstrap equations like 10 of the same CFT could be integrated.

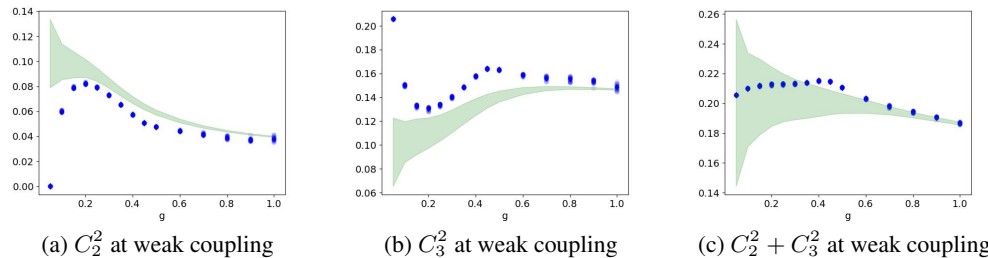

(a) $C_2^2$ at weak coupling     (b) $C_3^2$ at weak coupling     (c) $C_2^2 + C_3^2$ at weak coupling

Figure 1: Results on $C_2^2, C_3^2$ at weak coupling. Green regions represent the bounds from Cavaglià et al. (2022). The blue dots are the results for the best 25 runs based on reward for some values of $g$. Results have high precision, with a standard error of around $0.1\%$.

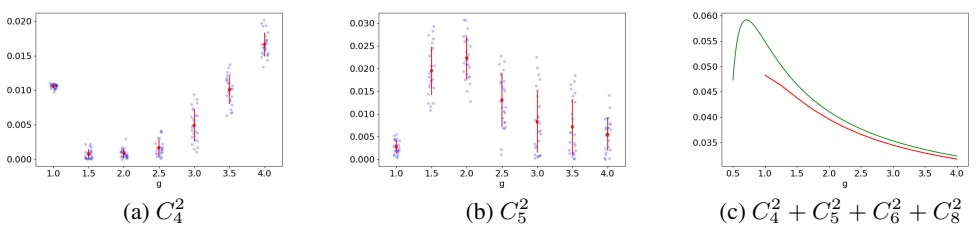

(a) $C_4^2$        (b) $C_5^2$        (c) $C_4^2 + C_5^2 + C_6^2 + C_8^2$

Figure 2: Strong coupling results for $C_4^2, C_5^2$ and $C_4^2 + C_5^2 + C_6^2 + C_8^2$ as a function of the coupling constant $g$. Blue points represent the individual values for the best 25 runs based on reward, while red points, lines and bars represent their means with standard deviation. Green lines represent theoretical expectations. Standard error on $C_4^2, C_5^2$ is around $10 - 50\%$ or worse and increases with $g$ due to the degeneracy problem. Standard error on $C_4^2 + C_5^2 + C_6^2 + C_8^2$ is calculated on the values of the sum for each run. Standard error is always below $0.1\%$, making the vertical bars invisible in figure 2c.

## 3.2 STRONG COUPLING

Since at strong coupling all bounds are tight enough, the middle points for $C_1^2, C_2^2, C_3^2$ are given as input to reduce the search space. In this case, we expect a higher precision since our experiments show that the best and average rewards increase monotonically with $g$ and the results are stable with multiple runs in parallel (see appendix A.4 for further results). Figures 2a and 2b show the results for $C_4^2, C_5^2$ at strong coupling. We can see that we have a higher precision on $C_4^2$, which is in line with the fact that the terms with higher $\Delta$ have a greater influence in the equation and are more optimized. However, the standard error for the best 25 runs seems to increase as $g \to 4$. Again, this is due to degeneracy as the scaling dimensions $\Delta_4, \Delta_5$ (together with $\Delta_6, \Delta_8$) converge to the same value of $\Delta = 6$ at infinity. Investigating on the sum $C_4^2 + C_5^2 + C_6^2 + C_8^2$ we found that our mean values are very close to a theoretical result from Ferrero & Meneghelli (2021); Ferrero & Meneghelli (2023); Ferrero & Meneghelli (2023), which indicates that the approach is working well (see figure 2c). We also noticed that if the standard deviation on the sum $C_4^2 + C_5^2 + C_6^2 + C_8^2$ is calculated using error propagation from the errors on the individual coefficients we obtain figure 6, with error between $1\%$ and $10\%$. If we instead calculate the sum $C_4^2 + C_5^2 + C_6^2 + C_8^2$ for each experiment and calculate statistics directly on these values, the results are surprisingly precise with errors always below $0.1\%$ and the bars of one standard deviation being invisible as in figure 2c. This is another confirmation of the effects of degeneracy: when two or more elements of the sum have similar values of $\Delta_n$, the sums of the corresponding coefficients are more precise than individual values.

## 4 CONCLUSION

In this work, we developed MultiSTOP, an extension to the BootSTOP algorithm that is able impose additional physical constraints into the framework. We experimented on a one-dimensional CFT

where some values of $\Delta_n$ and $C_n^2$ where known with enough precision to be given as input. At the weak coupling regime, the individual coefficients found were outside of the expected bounds due to the degeneracy problem. We found a similar issue at strong coupling, where the precision on results decreases with $g$. In both cases, the sum is much more aligned with previous results and has a higher precision. To solve this issue, we could include more equations into the same framework to decouple the coefficients or provide additional physical constraints and information. In the future, MultiSTOP can be applied on PDEs by using linear combinations of parametrized basis functions.

ACKNOWLEDGMENTS

This work has been supported by EU-EIC EMERGE (Grant No. 101070918).

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

## A  DETAILS ON THE MODEL AND THE ALGORITHM

### A.1  DETAILS ON THE MODEL'S EQUATIONS AND CONSTRAINTS

We start by defining the hypergeometric function $_2F_1(a, b, c; z)$ as

$$_2F_1(a, b, c; z) = \sum_{n=0}^{\infty} \frac{(a)_n (b)_n}{(c)_n} \frac{z^n}{n!}, \tag{5}$$

solutions to the Euler's hypergeometric differential equation

$$z(1 - z)\frac{d^2 f}{dz^2} + [c - (a + b + 1)z]\frac{df}{dz} - abf = 0, \tag{6}$$

where we have introduced the symbol $(q)_n$ as

$$(q)_n = \begin{cases} 1, & n = 0, \\ q(q+1)\cdots(q+n-1), & n > 0 \end{cases} \tag{7}$$

We also introduce the Bessel functions of the first kind $I_\alpha$ as

$$I_\alpha(x) = \sum_{m=0}^{\infty} \frac{1}{m!\Gamma(m+\alpha+1)} \left(\frac{x}{2}\right)^{2m+\alpha} \tag{8}$$

solutions to the differential equation

$$x^2\frac{d^2f}{dx^2} + x\frac{df}{dx} + (x^2 - \alpha^2) = 0. \tag{9}$$

Starting from Cavaglià et al. (2022), we now describe how the conformal bootstrap equation is expressed and implemented within our framework. To define equation 1 we begin with

$$x^2 f(1-x) + (1-x)^2 f(x) = 0 \tag{10}$$

where

$$f(x) = f_\mathcal{I}(x) + C_{BPS}^2 f_{\mathcal{B}_2}(x) + \sum_{n=1}^{\infty} C_n^2 f_{\Delta_n}(x). \tag{11}$$

The individual terms of $g$ defined as

$$\begin{aligned} f_\mathcal{I}(x) &= x \\ f_{\mathcal{B}_2}(x) &= x - x \,_2F_1(1, 2, 4; x) \\ f_\Delta(x) &= \frac{x^{\Delta+1}}{1-\Delta} \,_2F_1(\Delta+1, \Delta+2, 2\Delta+4; x) \end{aligned} \tag{12}$$

while the constant $C_{BPS}^2$ is given by

$$\begin{aligned} \mathbb{F}(g) &= \frac{3I_1(4\pi g)((2\pi^2 g^2 + 1)I_1(4\pi g) - 2g\pi I_0(4\pi g))}{2g^2\pi^2 I_2(4\pi g)^2} \\ C_{BPS}^2(g) &= \mathbb{F}(g) - 1 \end{aligned} \tag{13}$$

Finally, the functions in equation 1 are defined as

$$\begin{aligned} F_{\Delta_n}(x) &= x^2 f_{\Delta_n}(1-x) + (1-x)^2 f_{\Delta_n}(x) \\ h(x) &= x^2(f_\mathcal{I}(1-x) + C_{BPS}^2 f_{\mathcal{B}_2}(1-x)) + (1-x)^2(f_\mathcal{I}(x) + C_{BPS}^2 f_{\mathcal{B}_2}(x)) \end{aligned} \tag{14}$$

The integral constraints are defined as

$$\begin{aligned} \text{Constraint 1:} \quad \mathbb{I}_1 &= \sum_n C_n^2 \text{Int}_1[f_{\Delta_n}] + \text{RHS}_1 = 0, \\ \text{Constraint 2:} \quad \mathbb{I}_2 &= \sum_n C_n^2 \text{Int}_2[f_{\Delta_n}] + \text{RHS}_2 = 0. \end{aligned} \tag{15}$$

where

$$\begin{aligned} \text{Int}_1[f_{\Delta_n}] &= -\int_0^{\frac{1}{2}} (x - 1 - x^2)\frac{f_{\Delta_n}}{x^2}\partial_x \log(x(1-x))\mathrm{d}x \\ \text{Int}_2[f_{\Delta_n}] &= \int_0^{\frac{1}{2}} \frac{f_{\Delta_n}}{x^2}(2x - 1)\mathrm{d}x \end{aligned} \tag{16}$$

and the values $\text{RHS}_i, i = 1, 2$ are expressed by

$$\begin{aligned} \text{RHS}_1 &= \frac{\mathbb{B}(g) - 3C(g)}{8\mathbb{B}(g)^2} + \left(7\log 2 - \frac{41}{8}\right)(\mathbb{F}(g) - 1) + \log 2 \\ \text{RHS}_2 &= \frac{1 - \mathbb{F}(g)}{6} + (2 - \mathbb{F}(g))\log 2 + 1 - \frac{C(g)}{4\mathbb{B}(g)^2} \end{aligned} \tag{17}$$

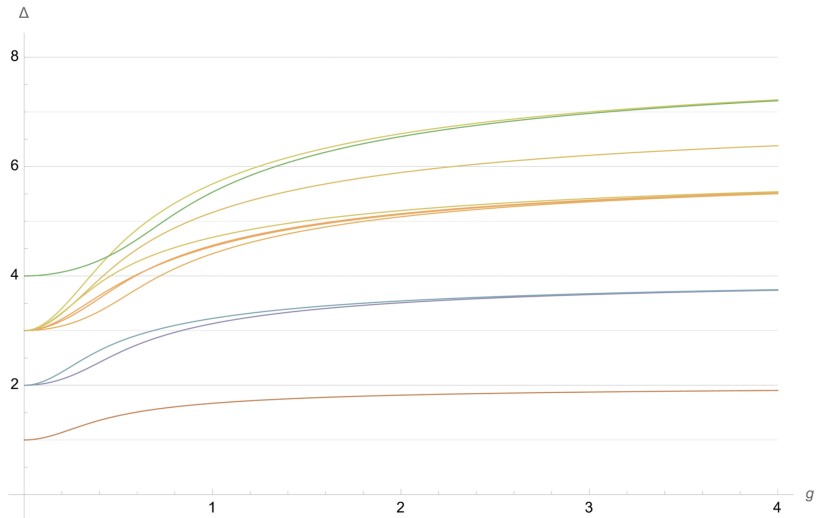

Figure 3: Values for the first 10 scaling dimensions $\Delta_n$ in the $1D$ CFT of interest, from Cavaglià et al. (2022).

where, finally,

$$\mathbb{B}(g) = \frac{g}{\pi} \frac{I_2(4\pi g)}{I_1(4\pi g)} \tag{18}$$

and the *curvature* $C(g)$ can be found in Cavaglià et al. (2022) as an expansion in terms of $g$.

Figure 3, taken from Cavaglià et al. (2022), shows the first 10 values of $\Delta_n$ as a function of the coupling constant $g$, which were used in our experiments.

## A.2 A SHORT RL INTRODUCTION

Reinforcement Learning is a particular Machine Learning technique involving two interacting components: an agent and the environment. At each time step, the agent has a state $S_t$, representing the stage of interaction, and does an action $A_t$ following a policy $\pi(A_t|S_t)$, which expresses the agent's behavior based on the state. Then, the environment outputs a reward signal $R_{t+1}$ and a new state $S_{t+1}$ is reached based on unknown dynamics $p(S_{t+1}, R_{t+1}|S_t, A_t)$. The objective of the agent is to find the optimal policy to maximize the expected discounted reward $\mathbb{E}_p[G_t]$ where $G_t = \sum_{i=t}^{\infty} \gamma^{i-t} R_i$ and $\gamma \in [0, 1]$ is the discount factor which expresses the importance of future rewards. RL techniques often involve studying objects such as the value function $v_\pi(s) = \mathbb{E}_{\pi,p}[G_t|S_t = s]$ and the action-value function $q_\pi(s, a) = \mathbb{E}_{\pi,p}[G_t|S_t = s, A_t = a]$ which are measures of the performance of the policy based on the current state and action.

## A.3 ALGORITHM DETAILS

The complete algorothm description of Bootstop can be found in algorithm 1.

### A.3.1 IMPLEMENTATION REMARKS

In order to improve the search and the precision of the final solution, two additional techniques are adopted. First, if the agent does not improve the maximum reward ever obtained after a certain number of steps, the Neural Networks of SAC are reset to start the search from scratch and try to find other possible values for the CFT data. Second, after some re-initializations of the networks, the search window is reduced around the previous best guess by a vertain factor to improve the precision. These two processes are repeated iteratively until enough precision is reached.

In our experiments, to further improve the accuracy and the performance of the agent, we fixed the numerical values for some parameters. In particular, the scaling dimensions $\Delta_n, 1 \leq n \leq 10$ are

---

**Algorithm 1** BootSTOP: Bootstrap STochastic OPtimization

---

Initialize parameters, best reward $R^* = 0$
**while** number of windows reductions less than `max_windows_exp` **do**
    **while** number of re-initializations less than `pc_max` **do**
        Initialize neural networks, agent and reset memory buffer.
        **for** Each time step $t$ **do**
            Agent selects action $(\Delta_n, C_n^2)$ with $n = t \bmod 10 + 1$.
            Update the state $S_t = (\boldsymbol{\Delta}, \boldsymbol{C^2})$.
            Calculate $F_{\Delta_n}(x)$ and crossing equations $O_t = \boldsymbol{E}(\boldsymbol{\Delta}, \boldsymbol{C^2})$.
            Calculate reward $R_t = \frac{1}{\|\boldsymbol{E}(\boldsymbol{\Delta}, \boldsymbol{C^2})\|}$.
            Agent receives reward $R_t$.
            Update memory buffer with the last transition.
            Update/learn parameters according to the main SAC algorithm
            **if** $R_t > R^*$ **then**
                Overwrite previous best reward $R^*$ and agent restart episode, $t = 0$.
            **end if**
            **if** number of steps without improving reaches `faff_max` **then**
                Exit For loop and reinitialize
            **end if**
        **end for**
    **end while**
    Reduce search windows size by a factor of `window_rate` centered around the state correspondent to $R^*$.
**end while**

---

known with high precision (Grabner et al., 2020) for all tested values of $g$ and are all given as input to the agent. Based on $g$, some values of the squared OPE coefficients $C_n^2$ are fixed beforehand as well, in particular the first (weak coupling case) or the first three (strong coupling case).

Similarly to Kàntor et al. (2023), we applied the speed-up techniques described in appendix A.3.2. In our case, the values of $F_{\Delta_n}(z_k, \bar{z}_k)$ are not approximated on a grid of selected values for the scaling dimensions $\Delta_n$ but are calculated on the accurate values from Grabner et al. (2020). Similarly, this lets us calculate beforehand the integrals in equation 2. This considerably increase the efficiency of the method and any arbitrary precision can be used to calculate these values without any cost when searching for the solution.

### A.3.2 SPEED-UP TECHNIQUES

When applying BootSTOP, the original authors noticed that the most time demanding part was the evaluations of the hypergeometric functions in the definition of the $F_{\Delta_n}(x)$ on the set of 180 points and the current guess for the CFT data. To alleviate this, they considered a grid of points for $\Delta_n$ with configurable precision. The values for $F_{\Delta_n}(x)$ on these values of $\Delta_n$ and $x$ are calculated beforehand and loaded into the algorithm for faster execution, at the cost of some precision.

### A.3.3 PARAMETERS

We report here the specific parameters used in the algorithm:

- Reward scale $r$, corresponding to the inverse of $\alpha$ in Haarnoja et al. (2018): 10. The $\tau$ parameter is set as 0.0005 and the discount factor is 0.99.
- The neural networks used in SAC are fully connected neural networks with 2 hidden layers of 256 units each. These are implemented and trained with PyTorch (Paszke et al., 2019) with the Adam optimizer and learning rate of 0.005.
- Number of steps without improvements in the maximum reward before re-initializing the networks: 10000.
- Number of re-inizializations before reducing the search windows: 10.
- Number of window reductions: 25 with constant reduction factor 0.7.

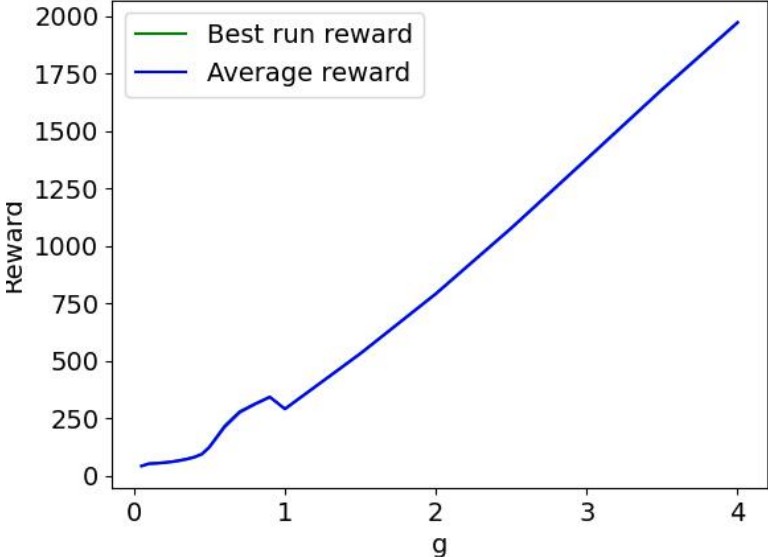

Figure 4: Experimental results on the values of the rewards as a function of the coupling constant $g$. The green line represents reward of the best performing run only, blue line and region represent the average of the best 25 runs and the standard deviation. The green line and blue region are almost invisible, showing the stability of the method.

- Number of parallel runs for each experiment: 500. Statistics are calculated on the best 25 runs in terms of the maximum reward obtained.

## A.4    FURTHER RESULTS

Figure 4 shows the best and averaged rewards on the best 25 runs as a function of the coupling constant $g$. As we can see, apart near $g = 1$ where the setting changes a bit from only $C_1^2$ to both $C_i^2, i = 1, 2, 3$ given as input, the reward monotonically increases. We also notice that the best reward is almost indistinguishable from the average of the top 25 with an almost null standard deviation. This shows the stability of the model under multiple runs. Figure 5 shows the experimental results on coefficient $C_6^2$ which have similar precision to the ones for $C_5^2$ in figure 2b. Figure 6 shows the results on the sum $C_4^2 + C_5^2 + C_6^2 + C_8^2$ with the standard deviation calculated with error propagation from the individual ones as described at the end of section 3.2.

## B    THEORETICAL BACKGROUND

### B.1    ON THE IMPORTANCE OF ADDING THE INTEGRAL CONSTRAINTS

We briefly describe the theoretical and experimental motivations for the addition of the integral constraints in equation 2. The conformal bootstrap, which leads to the main equation 10, is a powerful method that puts non-perturbative constraints on conformal field theory data $(\Delta, C^2)$. Previous approaches produced only allowed regions for the CFT data $(\Delta, C^2)$. The fact that the inclusion of the additional integral constraints in equation 2 significantly shrinks such regions, allowing in some cases to identify the theory under investigation with great precision, has been already observed in several papers in the physics literature, including the case that we study. We refer to Cavaglià et al. (2022); Chester et al. (2022); Chester (2023) for further details. It is therefore important to investigate whether the integral constraints are useful as it has been in many other contexts. A clear example of this is the improved precision between figures 6, 9, and 10 in Cavaglià et al. (2022). .

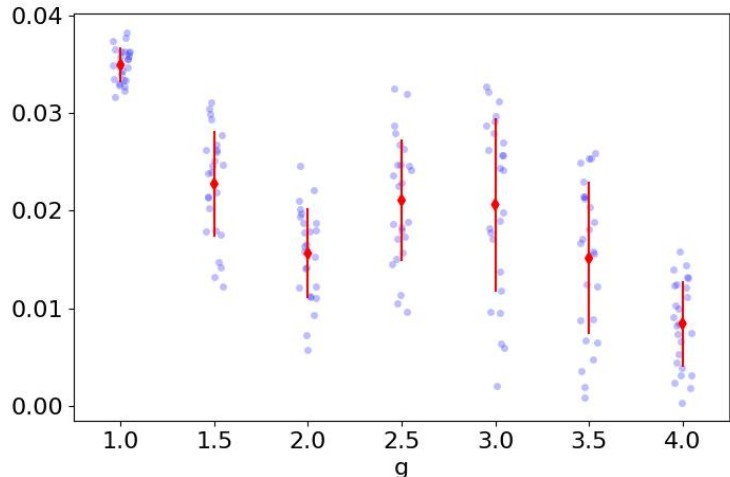

Figure 5: Strong coupling results for $C_6^2$ as a function of the coupling constant $g$. Blue points represent the individual values for the best $25$ runs based on reward, while red points and bars represent their means with standard deviation. Standard error is around $10 - 50\%$ or worse and increases with $g$ due to the degeneracy problem.

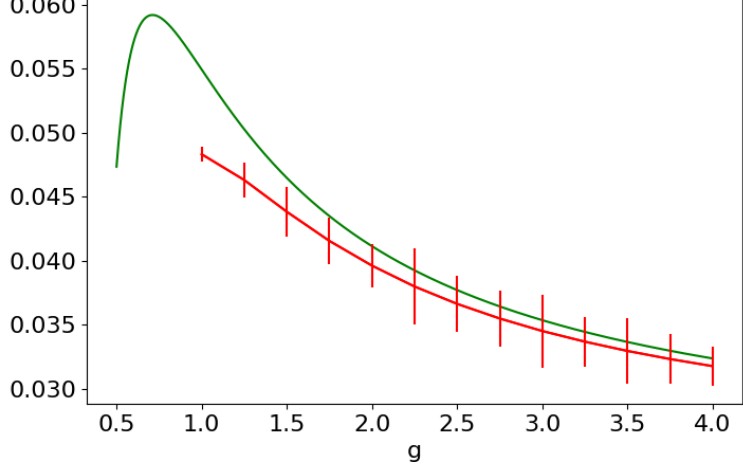

Figure 6: Experimental and theoretical results at strong coupling for the sum $C_4^2 + C_5^2 + C_6^2 + C_8^2$ as a function of the coupling constant $g$. Green lines represent the theoretical expectation and while red lines represent the mean values for the best $25$ rewards with their standard deviation. The total standard deviation is obtained via error propagation on the standard deviations on the individual coefficients $C_n^2$.

### B.2 DEGENERACY

On a theoretical level, individual components of the conformal bootstrap equation 10 represent physical states of the theory. We refer to *degeneracy* as the situation where two states $i$ and $j$ have conformal dimensions $\Delta_i$ and $\Delta_j$ (which we recall are functions of $g$) which become similar, to the point of being indistinguishable, for a certain value of $g$. This is the case at weak coupling ($g = 0$), and this degeneracy of states is responsible for the fact that the terms with $n = 2, 3$ can no longer be distinguished, so there is a significant loss in precision for the determination of their individual OPE coefficients $C_2^2$ and $C_3^2$. On the other hand, given the structure of the bootstrap equation, it is still possible to determine with good precision their sum $C_2^2 + C_3^2$. The fact that precision is lost at weak coupling is visible from the plots in figures 6, 9, 10 of Cavaglià et al. (2022), which focus on the first three states, while the OPE coefficients for other states are not shown. The fact that the degeneracy is particularly high at weak coupling (much higher than at strong coupling) can be seen from table 5 and figure 1 of Ferrero & Meneghelli (2023). While their terms in the equation are very similar, different states are physically different and can in theory be distinguished. This can be achieved by adding more conformal bootstrap equations of the same CFT into the MultiSTOP framework, which we leave for future works.

## C IMPROVED ACCURACY WITH MULTISTOP

To show the improvements of MultiSTOP compared to its baseline BootSTOP also on from an experimental point of view, we conducted experiments with $g = 1$ after the definitive weights were found to be $w_1 = 10^4, w_2 = 10^5$. We gave as input the conformal dimensions $\Delta_n$ and tried to match the known values for $C_1^2, C_2^2, C_3^2$, defined as the middle point of the available tight bounds from Cavaglià et al. (2022). In figure 7 we compare the most relevant metrics based on the number of integral constraints applied (the best 10 runs in each case are considered): from "no contstraints", the BootSTOP case, to "one constraint" or "two constraints" with MultiSTOP.

- The norm of the crossing equation 1 increases by a factor of two (figure 7a. This is most probably due to the harder optimization problem given by the addition of the integral constraints.

- Most importantly, the absolute relative error with respect to the ground truth (figures 7b,7c,7d) decreases when the integral constraints are applied, most notably when both of them are included. The reduction in error goes from a factor of at at least 2 for $C_2^2$ and $C_3^2$ to a factor of almost 10 in the case of $C_1^2$.

We can conclude that MultiSTOP is able to find more accurate solutions to the crossing equation when compared to the baseline method.

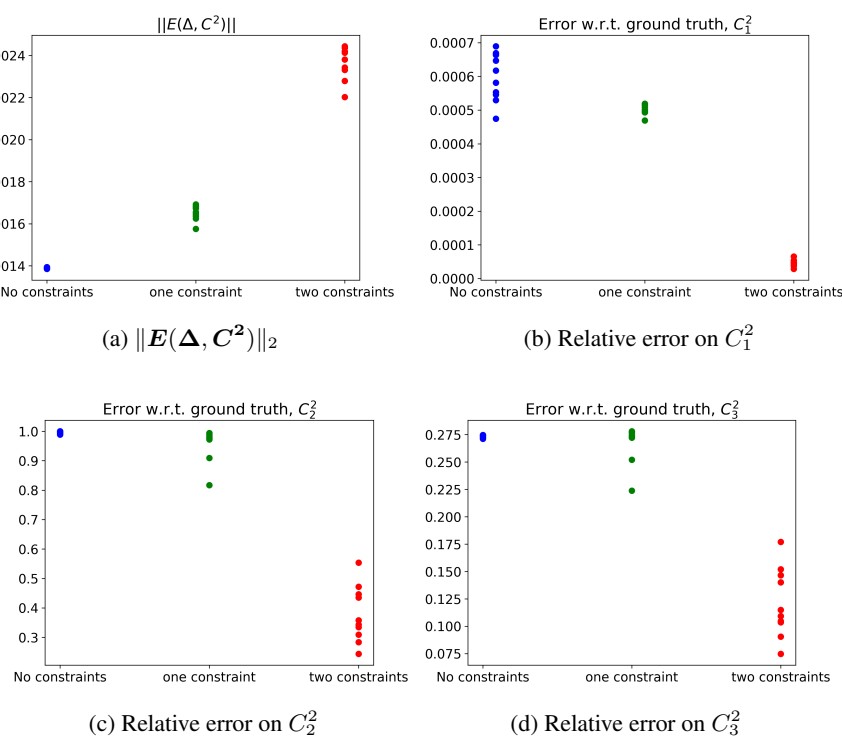

Figure 7: Comparisons between the baseline BootSTOP algorithm (no constraints applied) and the MultiSTOP methodology with one or two integral constraints applied. The norm of the crossing equation increases by less than a factor of 2, while the relative errors with respect to the known values for the first three coefficients is reduced by a factor between $2x$ and $10x$ on average with both constraints applied.

