# OpenReview forum: "MultiSTOP: Solving Functional Equations with Reinforcement Learning"
_ICLR.cc/2024/Workshop/AI4DiffEqtnsInSci — AI4DiffEqtnsInSci @ ICLR 2024 Poster_

### Official Review · Reviewer_oSM1 · 2024-02-19
**The authors present an extension of the BootStop algorithm, termed MULTIStop, by adding additional constrains derived from domain experts’ knowledge**

**Rating:** 5
**Confidence:** 2

**Review:**

The authors present an extension of the BootStop algorithm, termed MULTIStop, by adding additional constrains derived from domain experts’ knowledge. The authors use an 1D conformal bootstrap equation to show their method.

I think the paper is well written and make clear the main goals. I am not familiar with the example the authors use of the conformal bootstrap equation. I provide my comments below from an optimization perceptive and from a parameter non-identifiability angle of view.

(a)	The authors seem to be familiar with the BootStop Algorithm and here they propose their method MULTIStop which includes additional constrains to improve the accuracy of the bounds on their coefficients. I would expect to see at least a comparison with the BootStop algorithm to showcase that indeed the additional constrains are useful.

(b)	The authors choose to work in regimes for which there is literature suggesting what might be good OPE coefficients and scaling dimensions. I think this is the rational thing to do but it seems that when they use those “good initial conditions” the obtained coefficients C_n^2 are still off. Can they please comment on why this is the case?

(c)	The authors in many parts of the paper discuss about degeneracy; in my understanding this has to do with non-identifiability of the systems parameters (the coefficients that give a particular response cannot be determined uniquely –> the systems can reach the same state from different parameters) is this is case?

If yes, then in my opinion showing comparisons in terms of the inputs/parameters/coefficients space is misleading. I would expect so see how the norm (the output) ||E(Δ,C^2)||_2 looks like or how close to zero Equation 1 becomes for the obtained coefficients. If the values of their coefficients produce an error comparable to the prior work or smaller, then their method is successful. Based on the results showing in the manuscript is unclear to me the efficacy of their method.

(d)	As a follow up to my previous comment, I think this is why they observe much smaller errors when they use the sum C_4^2 + … + C_8^2. This might be an effective coefficient of the model and thus it has one-to-one correspondence with the output of interest.

In my opinion, the lack of a comparison between their approach and BootStop and the comparison only in the input space and not also in the output/response/behavior space makes the current paper marginally below acceptance threshold.

Minor Comment:
Abstract: Check the English of the second sentence.

---

### Meta-Review · Area_Chair_QMsk · 2024-03-01

**Recommendation:** Accept (Poster)

**Metareview:**

Thanks for the great points made by the reviewer. After going through the comments and paper I also think overall the paper is interesting. I expect the authors to address all the questions raised by reviewer in the final revision. Under this condition, this work can be accepted as poster.

---

### Decision · Program_Chairs · 2024-03-02

Accept (Poster)